# Unravelling the Proteomic Profiles of Bovine Colostrum and Mature Milk Derived from the First and Second Lactations

**DOI:** 10.3390/foods12224056

**Published:** 2023-11-07

**Authors:** Zhen Feng, Yan Shen, Gongjian Fan, Tingting Li, Caie Wu, Yuhui Ye

**Affiliations:** 1College of Light Industry and Food Engineering, Nanjing Forestry University, Nanjing 210037, China; fengzhen4158@dingtalk.com (Z.F.); fangongjian@njfu.edu.cn (G.F.); lttnjfu@njfu.edu.cn (T.L.); 2Co-Innovation Center for the Sustainable Forestry in Southern China, Nanjing Forestry University, Nanjing 210037, China; 3Jiangsu Key Laboratory for Food Quality and Safety—State Key Laboratory Cultivation Base of Ministry of Science and Technology, Institute of Food Safety and Nutrition, Jiangsu Academy of Agricultural Sciences, Nanjing 210014, China; shypaper@163.com

**Keywords:** bovine colostrum, lactation, mature milk proteomics, protein ingredient

## Abstract

Bovine colostrum (BC) and mature bovine milk are highly nutritious. In addition to being consumed by adults, these dairy products are also used as protein ingredients for infant formula. However, the differences in the nutritional composition of BC and mature milk, especially regarding proteins present in trace amounts, have not been comprehensively studied. Furthermore, the distinct proteomic profiles of mature milk derived from the first lactation (Milk-L1) and the second lactation (Milk-L2) are not fully understood. To address these gaps, this study aims to uncover the subtle differences in protein compositions of BC, Milk-L1, and Milk-L2 by proteomics. Compared with BC, anti-microbial proteins β-defensins and bovine hemoglobin subunit were up-regulated in Milk-L1, while Milk-L2 exhibited higher levels of enteric β-defensin, sterol regulatory element binding transcription factor 1, sydecan-2, and cysteine-rich secretory protein 2. Additionally, immune proteins such as vacuolar protein sorting-associated protein 4B, polymeric immunoglobulin receptor (PIGR), and Ig-like domain-containing protein were found at higher levels in Milk-L1 compared with Milk-L2. The study provides a comprehensive understanding of the distinct proteomic profiles of BC, Milk-L1, and Milk-L2, which contributes to the development of protein ingredients for infant formula.

## 1. Introduction

Bovine colostrum, the initial milk produced by cows in the first three days after parturition, is known to be rich in proteins, growth factors, immunoglobulins, and other bioactive compounds [1]. As lactation progresses, a transition from colostrum to mature milk occurs, during which the protein composition is significantly changed. Mature milk, in general, lacks the high levels of immunoregulatory compounds which are present in bovine colostrum, including immunoglobulins, hormones, growth factors, and cytokine [2]. However, it contains relatively high levels of casein, which can be further hydrolyzed, releasing casein-derived bioactive peptides and various essential amino acids [3]. Owing to the potent bioactivities of bovine colostrum and mature milk, they are considered important protein ingredients in the manufacturing of infant formula. Skim milk powder, processed from mature milk, is typically used as the casein source for infant formula, and by mixing it with whey protein concentrate (WPC), infant formula can mimic the casein-to-whey ratio and major nutrients present in human milk [4]. Furthermore, the supplementation of bovine colostrum to infants improves the digestive and absorptive functions, and ameliorates gut inflammation via the recovery of systemic immunity [5]. Although several studies have investigated the proteomic profiles of bovine colostrum [6,7,8] and mature milk [9,10] individually, there is a dearth of research into the comparison of these profiles to reveal the delicate differences in proteins enriched in each milk type. Therefore, it is of great importance to uncover and compare the protein profiles of bovine colostrum and mature milk, which may provide a better understanding of the nutritional values of infant formula, and also contribute to the development of infant formula with superior nutritional quality.

With regard to mature milk, the protein compositions can vary a lot between the first and second parities, which directly affects their nutritional values [11]. It has been reported that the total protein yield is increased in the second lactation of Holstein dairy cows, potentially owing to the development of the mammary gland and the regulation of hormones [12]. However, little is known about the specific changes in protein compositions in these two types of mature milk. Existing reports have found that the casein-to-whey ratios could shift during lactations, leading to a lower level of casein in the mature milk from the first lactation compared with the second lactation [13]. Furthermore, the colostrum proteins, including immunoglobulins, lactoferrin, and growth factors, may be more abundant in the mature milk from the first lactation [14]. Nevertheless, detailed information is lacking regarding the presence of other potentially functional proteins beyond those that have been extensively reported, such as casein, β-lactoglobulin, immunoglobulins, and lactoferrin. Hence, a systematic profiling of proteins is needed to elaborate on the subtle differences between mature milk from the first and second lactations, thereby shedding light on how each protein changes during lactations.

Nowadays, mass spectrometry-based proteomics are widely adopted in various research fields, such as clinical research, disease treatment, drug characterization, and nutrition [15]. As a high-throughput and high-sensitivity technique, proteomics allows for a comprehensive comparison of protein profiles among samples with the aid of multivariate statistical analyses. In the present study, a proteomics technique was employed, considering that various cytokines, growth factors, or other bioactive compounds with relatively low concentrations may be found inside bovine colostrum and mature milk. Therefore, this study aimed to uncover and compare the proteomic profiles of bovine colostrum and mature milk from the first and second lactations via mass spectrometry-based proteomics, thereby screening out the differentially expressed functional proteins and figuring out how they vary during lactations. This study provides valuable insights into the selection of ideal protein ingredients for infant formula.

## 2. Materials and Methods

### 2.1. Sample Information

The samples were obtained from the Experimental Farm of Yangzhou University, Jiangsu, China. Bovine colostrum (BC) was collected during the first 3 days, from the 2nd to 4th of September, 2020, after the cow had given birth to her first calf. Mature milk from the first lactation (Milk-L1) was obtained 20 days after the birth of the first calf, on the 18th of September, 2020. Mature milk from the second lactation (Milk-L2) was collected 20 days after the birth of the second calf, on the 3rd of December, 2021. Protein contents of the samples were measured immediately after collection by the experimental farm: BC 10% (*m*/*v*), Milk-L1 3.5% (*m*/*v*), Milk-L2 3.5% (*m*/*v*). After that, the milk samples were collected in 9 500 mL plastic bottles and stored in an ice-filled foam box for transportation. All the samples were transported to the laboratory within 6 h. Upon arrival, samples were allocated into 50 mL falcon tubes and stored at −60 °C.

### 2.2. The Protein Composition Visualized Using SDS-PAGE 

Sodium dodecyl sulphate-polyacrylamide gel electrophoresis (SDS-PAGE) was performed on all samples in triplicate. Samples were diluted 10 times with LDS sample buffer (4x, Invitrogen, Thermo, Waltham, MA, USA) to avoid overloading and heated at 80 °C for 10 min. Five µL of each sample was loaded in each well of a precast 12% Bis-Tris gel (NuPAGE, Thermo, USA). Proteins were separated at 150 V with NuPAGE MES SDS running buffer (Thermo, USA) for 1 h. Proteins were stained with 0.2% Coomassie Brilliant Blue R-250 solution in 85% phosphoric acid, ammonium sulphate (150 mg/mL), and 96% ethanol. The gels were destained in Milli-Q water and scanned using an Epson Perfection V750 Pro scanner (Seiko Epson Corporation, Nagano, Japan).

### 2.3. Mass Spectrometry-Based Proteomic Analysis

The proteins from 3 biological replicates of BC, Milk-L1, and Milk-L2 (100 μL, respectively) were extracted by 400 μL of cold acetone and thereafter redissolved in 8 M urea/100 mM Tris-Cl (pH 8.0). The solution was centrifuged (12,000× *g*, 4 °C) for 15 min and the supernatant was obtained. Ten mM of dithiothreitol was added and the solution was incubated at 37 °C for 1 h. Afterwards, 40 mM of iodoacetamide was added to permanently block the disulfide bond. The protein contents of samples were detected based on the Bradford method [16] and then normalized to 1 mg/mL. Trypsin (V5071, Promega, Madison, WI, USA) was added at a ratio of 50:1 (*w*/*w*) and the solution was incubated at 37 °C overnight. The pH of the solution was adjusted to 6.0 with trifluoroacetic acid to terminate the trypsin digestion. Finally, the supernatant was centrifuged again (12,000 rpm, 4 °C) for 3 min, and the newly obtained supernatant was collected for LC-MS/MS analysis, where the LC-ESI-MS/MS system (UPLC, EASY-nLC 1200, Thermo Scientific, MA, USA; MS, Orbitrap Exploris 480, Thermo Scientific, MA, USA) was applied. The analytical conditions were as follows: UPLC column, 50 μm × 150 mm, 2 μm particle size, 100 Å pore size, Acclaim PepMap C18 column, Thermo; column temperature, 40 °C; flow rate, 0.4 mL/min; injection volume, 5 μL; solvent system, water (0.1% formic acid): acetonitrile (0.1% formic acid); gradient program, 95:5 *v*/*v* at 0 min, 10:90 *v*/*v* at 10.0 min, 10:90 *v*/*v* at 11.0 min, 95:5 *v*/*v* at 11.1 min, 95:5 *v*/*v* at 14.0 min. The ESI source operation parameters were as follows: source temperature, 500 °C; ion spray voltage (IS), 5500 V (positive) or −4500 V (negative); ion source gas I, gas II and curtain gas were set at 50, 50, and 25 Psi, respectively; the collision gas was set as high. The raw data were extracted by DIA-Umpire software (v2.2.8) [17] and MSFragger (v3.8) [18] was used for matching the data to a known bovine protein database, Uniprot. Proteome. Bovine. 20200629. fasta. All the settings of MSFragger were set to default except for the variable modifications, which were ‘Oxidation (M), Acetyl (Protein N-term)’, and the fixed modification was ‘Carbamidomethyl (C)’ and the digestion enzyme was ‘Trypsin’. 

### 2.4. Multivariate Statistical Analyses

Protein intensities were log_2_-transformed for statistical analysis to improve normality. All the proteins were used for hierarchical clustering analysis (HCA) with Perseus [19]. Unsupervised PCA (principal component analysis) was performed using statistics function prcomp within R (www.r-project.org, accessed on 20 March 2023). The R package was also used for the hierarchical cluster analysis. Differentially expressed proteins were selected based on absolute Log_2_FC (fold change: the ion intensity of the protein in experimental group/the ion intensity of the protein in control group) > 2 and *p* < 0.05. The differentially expressed proteins were then annotated and enriched using the Kyoto encyclopedia gene and genome (KEGG) compound database (http://www.kegg.jp/kegg/compound/, accessed on 20 March 2023).

## 3. Results

### 3.1. The Distribution of Proteins in Bovine Colostrum and Mature Milk from the First and Second Lactations

SDS-PAGE was performed in BC, Milk-L1, and Milk-L2 (Figure 1). High molecular weight protein aggregates (bands > 180 kDa) were observed in BC and Milk-L1, but were absent in Milk-L2. The streaking region (75–100 kDa) which corresponded to lactoferrin (~80 kDa)/lactoperoxidase (~78 kDa) was observed in BC and Milk-L1, with the band intensity in BC higher than that in Milk-L1. The bands 60–75 kDa, which may refer to BSA (bovine serum albumin, ~66 kDa), were observed in all the samples. The bands with a molecular weight slightly below 60 kDa could be the heavy chain of immunoglobulin G (~55 kDa), and the band intensity followed the pattern BC > Milk-L1 > Milk-L2. Additionally, caseins were separated in bands within the range between 25 and 35 kDa, and whey proteins, including β-lactoglobulin (18.4 kDa) and α-lactalbumin (14 kDa), were observed in all the samples. The band intensities of caseins followed the pattern Milk-L2 > BC > Milk-L1.

### 3.2. Multivariate Statistical Analyses of Bovine Colostrum and Mature Milk from the First and Second Lactations

Multivariate statistical analyses were performed to assess the differences between the proteomic profiles of BC, Milk-L1, and Milk-L2. Principal component analysis (PCA) was firstly applied to investigate the internal structure of multiple variables according to different principal components. The result indicated that the three types of milk were divided into three distinct groups with principal component 1 (PC1) at 58.05% and principal component 2 (PC2) at 27.96% (Figure 2A). BC was clearly separated by PC1, while Milk-L1 was discriminated by PC2. 

In total, 749 proteins were detected by mass spectrometry in the present study. The overall proteomic differences of different groups were further visualized by hierarchical clustering analysis, where three major clusters were obtained (Figure 2B). The result indicated that cluster 3 contained the most proteins, where the average intensity of the heat map followed the pattern BC < Milk-L1 < Milk-L2.

### 3.3. Comparison of the Proteomic Profiles of Bovine Colostrum and Mature Milk from the First Lactation

There were 458 up-regulated and 13 down-regulated proteins in Milk-L1 compared with BC (Figure 3A). Among these, the most up-regulated proteins included procollagen galactosyltransferase 1, hemoglobin subunit β, α-actinin-4, β-defensin 8, and enteric β-defensin. However, Ig-like domain-containing protein, transcription factor E2F6, pentraxin-related protein PTX3, and inter-α-trypsin inhibitor heavy chain H3 were down-regulated, and thus showed higher levels in BC (Figure 3B). All the differentially expressed proteins between BC and Milk-L1 were then mapped to the KEGG database to reveal the related biological pathways. The differentially expressed proteins were mainly involved in the pathways of human diseases, including shigellosis, systemic lupus erythematosus, and pathogenic *E. coli* infection (Figure 3C).

### 3.4. Comparison of the Proteomic Profiles of Bovine Colostrum and Mature Milk from the Second Lactation

In the comparison of Milk-L2 and BC (control), 458 proteins were found to be up-regulated, while 13 proteins were down-regulated (Figure 4A). According to the fold change, the most up-regulated proteins were SEA domain-containing protein, sterol regulatory element binding transcription factor 1, syndecan-2, cysteine-rich secretory protein 2, and enteric β-defensin. However, vacuolar protein sorting-associated protein 4B, PIGR, kazal-like domain containing protein, and ephrin-A1 were among the most down-regulated proteins (Figure 4B). The differentially expressed proteins were also mostly involved in the pathways of human diseases, including salmonella infection and pathogenic *E. coli* infection (Figure 4C).

### 3.5. Comparison of the Proteomic Profiles of Mature Milk from the First and Second Lactations

Regarding the comparison between Milk-L1 (control) and Milk-L2, 172 proteins were down-regulated, whereas 169 proteins were up-regulated (Figure 5A). Milk-L1 exhibited higher levels of vacuolar protein sorting-associated protein 4B, PIGR, fatty acid-binding protein 5, Ig-like domain-containing protein, and heterogeneous nuclear ribonucleoprotein A1, while Milk-L2 showed higher levels of sterol regulatory element binding transcription factor 1, transcription factor E2F6, syndecan-2, zinc finger protein CCCH-type, and PHD finger protein 3 (Figure 5B). The KEGG analysis revealed that the differentially expressed proteins were mainly involved in the ECM (extracellular matrix)-receptor interaction and the AMPK signaling pathways (Figure 5C).

## 4. Discussion

Bovine colostrum is increasingly being used as a nutritional supplement to promote gut function and health in human newborns, owing to its high levels of immunoregulatory proteins. Mature milk, both bovine and goat milk, also plays significant roles in infant nutrition and is extensively used as the protein base for infant formula. Therefore, a comprehensive comparison between the proteomic profiles of bovine colostrum and mature milk is needed to shed light on the specific nutrition that can be provided to infants. Additionally, the biological pathways annotated by the differentially regulated proteins may indicate the potential biological functions of bovine colostrum and mature milk. 

SDS-PAGE was first conducted in order to reveal the distribution of the major proteins present in bovine colostrum and mature milk from the first and second lactations. According to the results of protein separation, some protein aggregates and streaking regions were observed in BC and Milk-L1. This could have resulted from improper storage during transportation, where heat load induced the aggregation of milk proteins through either covalent binding or hydrophobic interactions [20]. However, these aggregated proteins had no effect on the following proteomic analysis due to the urea, dithiothreitol, and iodoacetamide pre-treatments. BC contained more IgG (in both aggregated form and heavy chain form) and lactoferrin than Milk-L1, while Milk-L2 had limited amounts of these large proteins. This result was in line with the findings of Goldsmith et al., who indicated that total immunoglobulins and lactoferrin were decreased significantly with the increase in postpartum time [21]. After birth, newborn calves are exposed to a massive invasion of potentially harmful pathogens from the environment and the immunoglobulins provided by bovine colostrum play important roles in the calves’ innate immune defense. Moreover, IgG, the major type of immunoglobulin in bovine colostrum, has proven to be a promising supplement in specialized dairy products, supporting immune function in vulnerable groups such as infants, children, the elderly, and immunocompromised patients [22]. The casein-to-whey ratio was significantly changed during lactations, as Milk-L2 had a higher casein-to-whey ratio than Milk-L1, which was in line with previous findings that the mammary gland is more mature and efficient at milk synthesis during the second lactation [13,23,24]. 

Following the general protein differences uncovered by SDS-PAGE, proteomics was conducted to analyze the delicate differences between the protein profiles of the different milk types. During the transition from BC to Milk-L1, proteins were mostly up-regulated, including procollagen galactosyltransferase 1, hemoglobin subunit β, α-actinin-4, β-defensin 8, and enteric β-defensin. The result revealed that enteric β-defensins were significantly accumulated in mature milk, relative to bovine colostrum. β-Defensins are antimicrobial peptides primarily expressed by leukocytes and epithelial cells at mucosal surfaces. In bovine milk, β-defensins act as a part of the bovine innate immune system and the first line of defense against bovine mastitis [25]. The significant antimicrobial activity of β-defensins can facilitate both innate and adaptive immune responses in humans [26]. In addition, the bovine hemoglobin subunit which accumulated in Milk-L1 also exhibited potent antimicrobial activity in humans. Although the entire hemoglobin molecule showed no direct antibacterial activity, its fragments, the hemoglobin derived peptides, were bioactive as they not only exhibited an inhibitory effect on *E. coli* and *S. aureus*, but also took part in wound healing processes at injured sites [27]. Therefore, the anti-microbial proteins were up-regulated in Milk-L1 compared with BC. However, BC contained elevated levels of several anti-inflammatory proteins, including pentraxin-related protein PTX3 and inter-α-trypsin inhibitor heavy chain H3. PTX3, a pattern recognition molecule involved in innate immune responses, is able to cooperate with the adaptive immune system in defense inflammation via developing a protective Th1/T_reg_ immune response and inhibiting a harmful Th17/Th2 immune response [28,29]. Similarly, inter-α-trypsin inhibitor heavy chain H3 is a plasma protein associated with inflammation and tumorigenic and metastatic processes. Normally, this protein is linked to hyaluronic acid to maintain extracellular matrix stability. The dysregulation of inter-α-trypsin inhibitor protein can lead to inflammation and promote tumor development [30]. In addition, Ig-like domain-containing protein, which regulates the immune response against inflammation, was unsurprisingly more abundant in BC. Additionally, the result of KEGG annotation further confirmed that the anti-microbial related pathways, such as shigellosis and pathogenic *E. coli* infection, were enriched during the transition from BC to Milk-L1. 

Regarding the comparison between Milk-L2 and BC, the anti-microbial protein enteric β-defensin also accumulated in mature milk from the second lactation, which was in accordance with the result from the comparison between Milk-L1 and BC. Moreover, sterol regulatory element binding transcription factor 1, the key transcription factor for the nutritional induction of lipogenic enzyme genes, exhibited a significantly higher level in Milk-L2 compared with BC. A similar result was observed in [31], which indicated that the sterol regulatory element binding transcription factor 1 in mature milk facilitates a healthier fatty acid composition by modulating the amount of lauric acid (12:0) and myristic acid (14:0). Milk-L2 also contained a significantly higher level of sydecan-2 compared with BC. Sydecan-2 plays important roles in neuronal development and functions as a tumor suppressor regulating the apoptotic signaling in bone tumors [32]. In addition, another functional protein, cysteine-rich secretory protein 2, was detected in Milk-L2. Cysteine-rich secretory proteins are key players in mammalian fertilization and fertility, having Ca^2+^ channel regulatory abilities [33]. To our knowledge, the present study is the first to report the detection of cysteine-rich secretory protein 2 in milk, despite this protein being predominately secreted by arthropods. In BC, immunoregulatory proteins PIGR and kazal-like domain-containing protein were dominant. PIGR is able to transport immunoglobulin A across the epithelial cell layer into mucosal and glandular surfaces of mammals, such as the gastrointestinal tract, respiratory tract, urogenital tract, and the mammary glands, thereby protecting against different infections [34]. Kazal-like domain containing protein is a member of the insulin growth factor-binding proteins (IGFBP) which regulate the bioavailability and function of insulin growth factors [35]. Like PIGR and immunoglobulins, insulin growth factors are important immunomodulatory proteins in BC, showing a significant protective effect for infant health. A human study confirmed a positive correlation between the body weight of a 1-year-old infant and concentrations of insulin growth factor-I in the milk from the diet [36]. In addition, compared with Milk-L2, BC also contained higher levels of vacuolar protein sorting-associated protein 4B and ephrin-A1, which both control the signaling transduction in tumor formation [37,38]. 

Mature milk is not only consumed by adults, but is also regarded as an important protein ingredient in infant formula. The skim milk powder processed from mature milk can balance the casein-to-whey ratio and provide various nutrients for infants. However, the protein composition of mature milk varied between parities. In comparison to Milk-L2, Milk-L1 contained higher levels of vacuolar protein sorting-associated protein 4B, PIGR, and Ig-like domain-containing protein, which were also abundant in BC. This indicated that the mature milk from the first lactation still had several immune proteins in common with BC. The result was in line with [39], which demonstrated that milk from the second lactation contains the lowest immunological composition. This could potentially be explained by the fact that first-lactation cows have a less pronounced negative energy balance and stable metabolic profiles, while second-lactation cows experience more immune reactions where proteins are consumed for the development of the mammary glands [12]. However, functional proteins sterol regulatory element binding transcription factor 1 (able to regulate fatty acid composition) and syndecan-2 (anti-tumor) were of higher levels in Milk-L2 compared in Milk-L1. In addition, a high level of zinc finger protein CCCH-type endowed Milk-L2 with an anti-inflammatory effect [40]. KEGG annotation revealed the major pathways that Milk-L2 was involved in. The enrichment of the ECM-receptor interaction pathway may indicate the occurrence of elevated immune reactions, while the detection of the AMPK signaling pathway provided evidence for energy consumption in Milk-L2. Therefore, in contrast to the increased levels of immune proteins in Milk-L1, Milk-L2 contained a wider range of proteins with diverse bioactivities, including anti-inflammation, anti-tumor, and fatty acid regulation. 

## 5. Conclusions

The proteomic profiles of bovine colostrum, and mature milk from the first and second lactations were revealed and compared in the present study. Our results indicated that BC contained more IgG and lactoferrin than Milk-L1, while Milk-L2 had the lowest levels of these proteins. However, Milk-L2 had a higher casein-to-whey ratio than Milk-L1 and BC. In comparison to BC, anti-microbial proteins enteric β-defensin, β-defensin 8 and bovine hemoglobin subunit were up-regulated in Milk-L1, while Milk-L2 exhibited higher levels of enteric β-defensin, sterol regulatory element binding transcription factor 1, sydecan-2, and cysteine-rich secretory protein 2. In addition, Milk-L1 contained increased levels of immune proteins, such as vacuolar protein sorting-associated protein 4B, PIGR, and Ig-like domain-containing protein, compared with Milk-L2. However, Milk-L2 exhibited a protein profile with diverse bioactivities, including anti-inflammation, anti-tumor, and fatty acid regulation properties. In summary, this study provides valuable insight into the delicate differences between the proteomic profiles of bovine colostrum and mature milk from the first and second lactations, thereby contributing to the development of ideal protein ingredients for infant formula.

## Figures and Tables

**Figure 1 foods-12-04056-f001:**
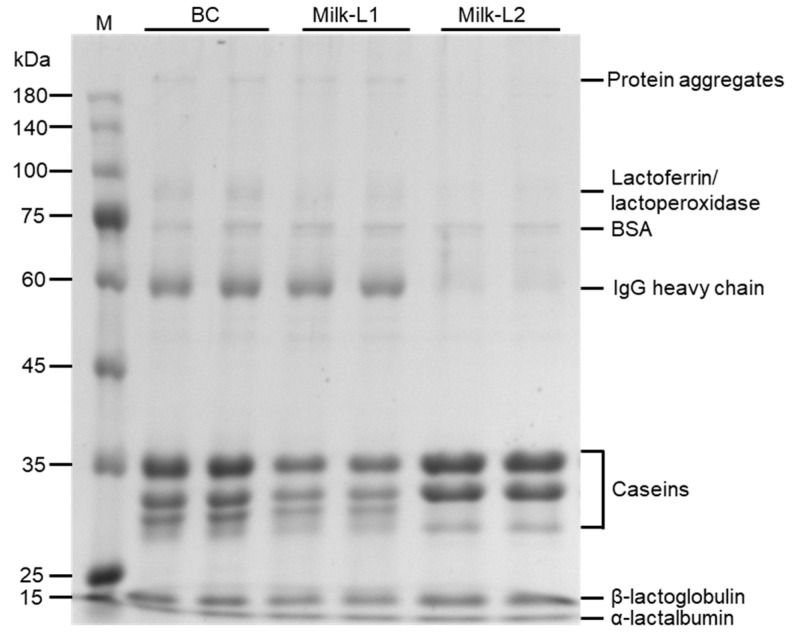
SDS-PAGE of proteins of bovine colostrum and milk from the first and second lactations; M: protein marker, BC: bovine colostrum, Milk-L1/2: mature milk from the first/second lactation.

**Figure 2 foods-12-04056-f002:**
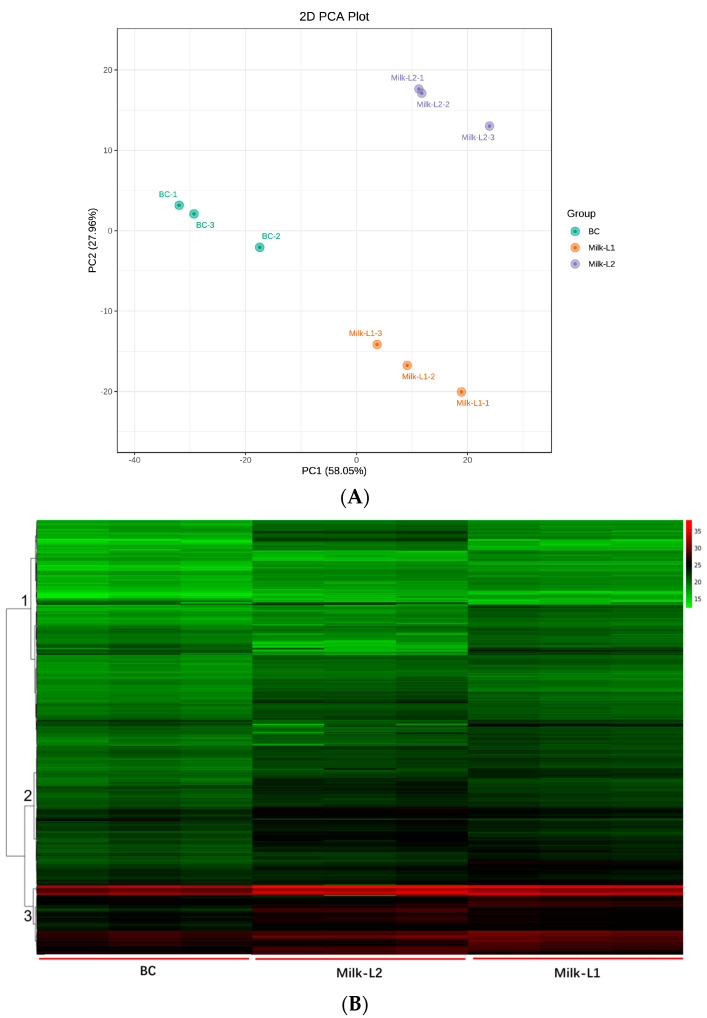
(**A**) PCA plot of bovine colostrum and mature milk from the first and second lactations; BC: bovine colostrum, Milk-L1/2: mature milk from the first/second lactation. (**B**) Hierarchical clustering of bovine colostrum and mature milk from the first and second lactations; BC: bovine colostrum, Milk-L1/2: mature milk from the first/second lactation. The colors green, black, and red indicated proteins with relatively low, middle, and high ion intensities, respectively.

**Figure 3 foods-12-04056-f003:**
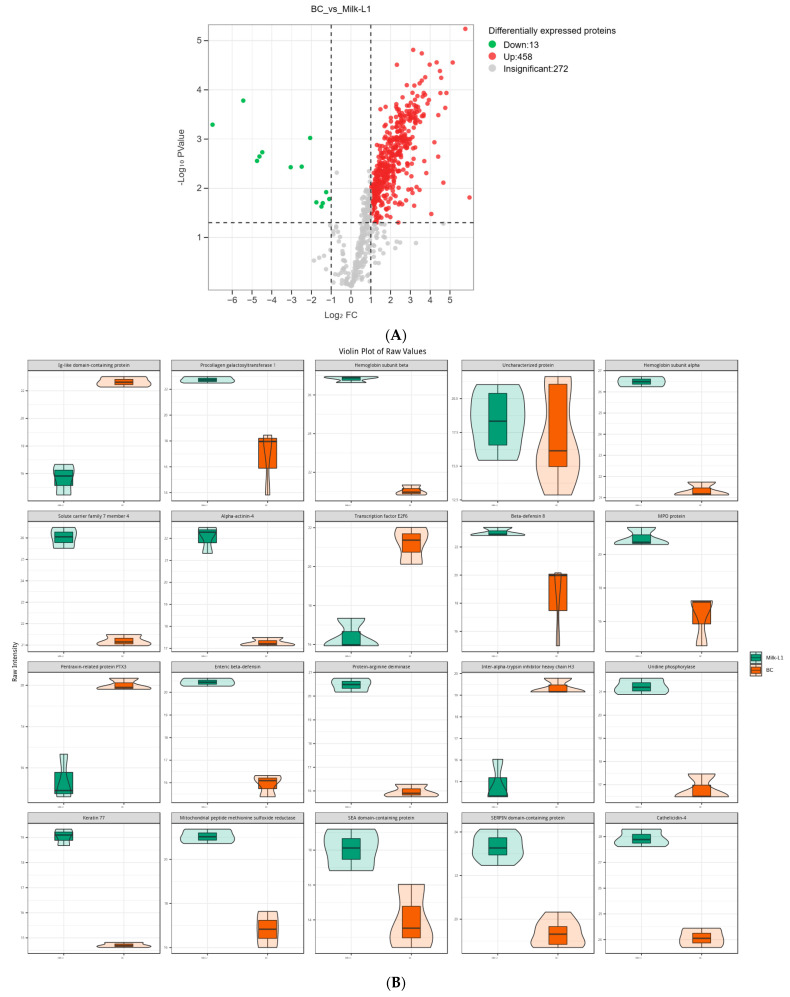
(**A**) Volcano plot of the differentially expressed proteins in bovine colostrum (control) and milk from the first lactation; BC: bovine colostrum, Milk-L1: mature milk from the first lactation. The red dots indicate the significantly up-regulated proteins, while the green dots show the significantly down-regulated proteins. The significance threshold is Log_2_FC (fold change) > 2 and *p* < 0.05, relative to control. (**B**) Representatives of the differentially expressed proteins regarding the comparison between bovine colostrum (control) and mature milk from the first lactation; BC: bovine colostrum, Milk-L1: mature milk from the first lactation. The box in the middle represents interquartile range. The 95% confidence interval is presented as a black line which penetrates through the box. The black line in the middle of the box is the median. The outer shape describes the density of distribution. (**C**) KEGG annotation of differentially expressed proteins in the comparison between bovine colostrum (control) and milk from the first lactation; the length of bars indicates the number of proteins annotated in each pathway. *p* value is marked at the end of each bar.

**Figure 4 foods-12-04056-f004:**
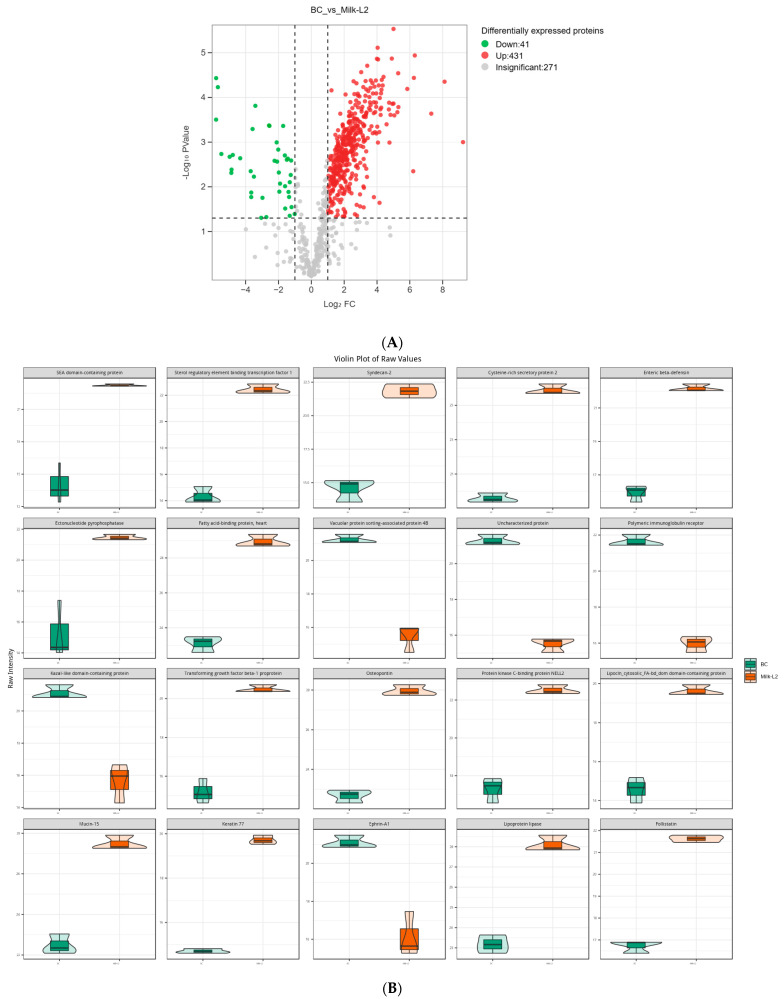
(**A**) Volcano plot of the differentially expressed proteins in bovine colostrum (control) and milk from the second lactation; BC: bovine colostrum, Milk-L2: mature milk from the second lactation. The red dots indicate the significantly up-regulated proteins, while the green dots show the significantly down-regulated proteins. The significance threshold is Log_2_FC (fold change) > 2 and *p* < 0.05, relative to control. (**B**) Representatives of the differentially expressed proteins regarding the comparison between bovine colostrum (control) and milk from the second lactation; BC: bovine colostrum, Milk-L2: mature milk from the second lactation. The box in the middle represents interquartile range. The 95% confidence interval is presented as a black line which penetrates through the box. The black line in the middle of the box is the median. The outer shape describes the density of distribution. (**C**) KEGG annotation of differentially expressed proteins in the comparison between bovine colostrum (control) and milk from the second lactation; The length of bars indicates the number of proteins annotated in each pathway. *p* value is marked at the end of each bar.

**Figure 5 foods-12-04056-f005:**
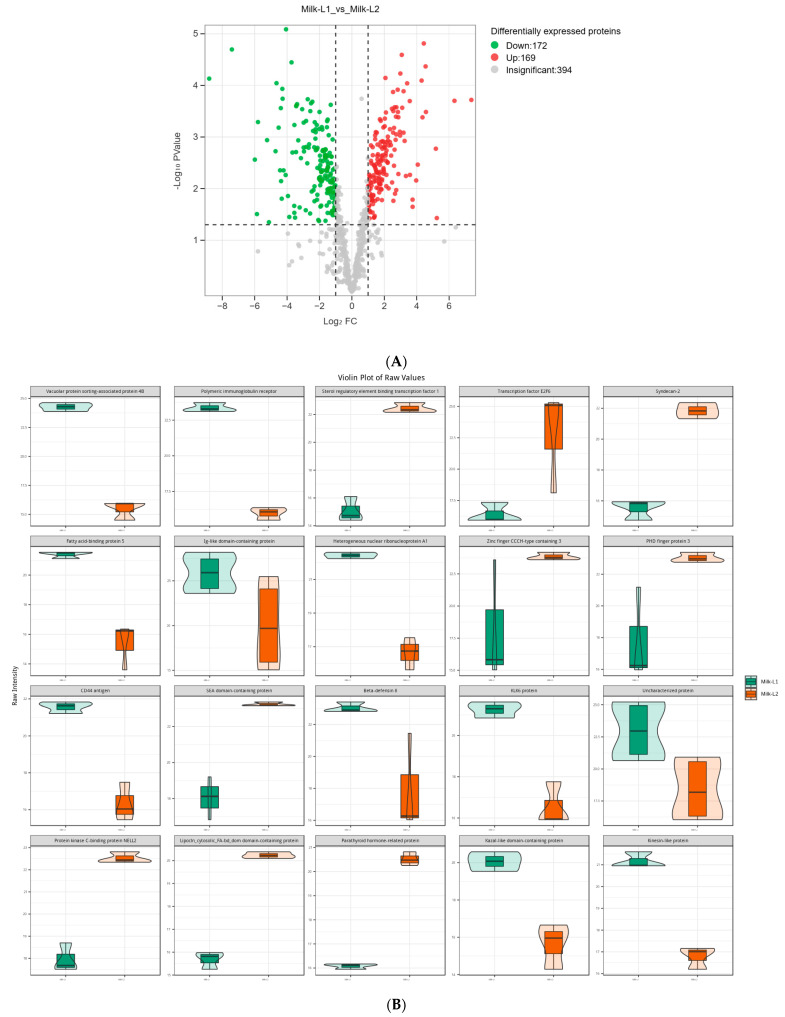
(**A**) Volcano plot of the differentially expressed proteins in mature milk from the first (control) and second lactations; Milk-L1/2: mature milk from the first/second lactation. The red dots indicate the significantly up-regulated proteins, while the green dots show the significantly down-regulated proteins. The significance threshold is Log_2_FC (fold change) > 2 and *p* < 0.05, relative to control. (**B**) Representatives of the differentially expressed proteins regarding the comparison between mature milk from the first (control) and second lactations; Milk-L1/2: mature milk from the first/second lactation. The box in the middle represents interquartile range. The 95% confidence interval is presented as a black line which penetrates through the box. The black line in the middle of the box is the median. The outer shape describes the density of distribution. (**C**) KEGG annotation of differentially expressed proteins in the comparison between mature milk from the first (control) and second lactations; The length of bars indicates the number of proteins annotated in each pathway. *p* value is marked at the end of each bar.

## Data Availability

The data used to support the findings of this study can be made available by the corresponding author upon request.

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
