# Peer review of "Unravelling the Proteomic Profiles of Bovine Colostrum and Mature Milk Derived from the First and Second Lactations"

_foods, 2023, doi:10.3390/foods12224056_

Round 1

Reviewer 1 Report

Comments and Suggestions for Authors

The manuscript “Unravelling the Proteomic Profiles of Bovine Colostrum and 2 Mature Milk Derived from the First and Second Lactations” makes a comparative analysis of the proteome shown by bovine colostrum and mature milk obtained from first and second lactation.

The originality of this study is not so high. However, the presentation is clear, methods appear appropriate and the conclusions are supported by the obtained data. Definetely, the results can be useful for improving the infant formula.

In general, I would suggest to check the text for language inaccuracies.

Some minor comments/suggestions

1. Materials and Methods, lines 83-86. “Mature milk from the first lactation (Milk-L1) was obtained 20 days after the birth of the first calf, at 18th of September, 2020. Mature milk from the second lactation (Milk-L2) was collected 20 days after the birth of the second calf, at 3rd of December, 2021.”

Was the cow diet in September the same as that in December?

2. Materials and Methods, Par. 2.3 “Mass spectrometry-based proteomic analysis” and elsewhere.

Please check the capitalized first letter of some words. Generally, after comma, semicolon and colon, the word begins with a lowercase letter.  

3. Results, line 141. “High molecular proteins…”

Should it be "...high molecular weight proteins ..."?

4. Results, line 164. “…749 proteins were detected in the present study.”

How were proteins detected? Mass spectrometry? Please specify in the text.

5. Figure 3B.

If possible, the size of words and numbers in Figure 3B could be increased.

6. Legend to Figure 3C.

Please specify that this Figure shows "differentially expressed proteins in BC compared with Milk-L1"

7. Discussion, Line 259. “…newborns…”

Do you mean "human newborns"?

8. Discussion, Line 316. “E. coli”

Scientific names should be in italic, as the Authors write elsewhere.

Comments on the Quality of English Language

I would suggest to check the text for language inaccuracies.

Author Response

Dear editor

Thank you so much for considering our work for publication in Foods. We appreciate all the reviewer’s comments and suggestions. All changes in the revised manuscript have been marked up using the “Track Changes” function, so it is more visible and easier to read. In addition, the specific changes and line number have been stated in each response. Here is a point-by-point response to the reviewer’s comments and concerns.

  1. Materials and Methods, lines 83-86. “Mature milk from the first lactation (Milk-L1) was obtained 20 days after the birth of the first calf, at 18th of September, 2020. Mature milk from the second lactation (Milk-L2) was collected 20 days after the birth of the second calf, at 3rd of December,2021.” Was the cow diet in September the same as that in December?

Response: Yes, the cow diets are the same between September and December. Therefore, the protein compositions of Milk-L1 and Milk-L2 are comparable. Thanks.

  1. Materials and Methods, Par. 2.3 “Mass spectrometry-based proteomic analysis” and elsewhere. Please check the capitalized first letter of some words. Generally, after comma, semicolon and colon, the word begins with a lowercase letter. 

Response: We have changed our subtitles to lowercase forms. For example, “Mass spectrometry-based proteomic analysis” has changed to “mass spectrometry-based proteomic analysis”. Please see Lines 80, 92, 102, 128, 139, 155, 175, 204 and 232. Thanks.

  1. Results, line 141. “High molecular proteins…” Should it be "...high molecular weight proteins ..."?

Response: We have changed “High molecular proteins…” to "High molecular weight proteins ..." Please see Line 142. Thanks.

  1. Results, line 164. “…749 proteins were detected in the present study.” How were proteins detected? Mass spectrometry? Please specify in the text.

Response: We have changed  “…749 proteins were detected in the present study.” to “…749 proteins were detected by mass spectrometry in the present study.” Please see Line 163. Thanks.

  1. Figure 3B. If possible, the size of words and numbers in Figure 3B could be increased.

Response: We will upload the figures with the highest resolution and make sure each word can be seen clearly when zoomed in. Thanks.

  1. Legend to Figure 3C. Please specify that this Figure shows "differentially expressed proteins in BC compared with Milk-L1"

Response: Figure 3C was unable to present the "differentially expressed proteins in BC compared with Milk-L1". However, it was the result of KEGG annotation of differentially expressed proteins. Therefore, we have slightly modified the legend of Figure 3C, from "KEGG analysis regarding the comparison between bovine colostrum (control) and milk from the first lactation" to "KEGG annotation of differentially expressed proteins in the comparison between bovine colostrum (control) and milk from the first lactation". Please see Lines 200, 228 and 256. Thanks.

  1. Discussion, Line 259. “…newborns…” Do you mean "human newborns"?

Response: Yes, therefore we have changed “…newborns…” to “…human newborns…” Please see Line 262. Thanks.

  1. Discussion, Line 316. “E. coli” Scientific names should be in italic, as the Authors write elsewhere.

Response: We have changed “E. coli” to “E. coli” as suggested. Please see Line 320. Thanks.

Best regards,

Yuhui Ye, PhD

Jiangsu Academy of Agricultural Sciences

Reviewer 2 Report

Comments and Suggestions for Authors

Informative and comprehensive study: The article provides a detailed and systematic analysis of the protein composition of bovine colostrum (BC) and mature bovine milk from the first and second lactations. The article uses advanced proteomics techniques to identify and quantify the proteins present in trace amounts in these dairy products, which have important implications for human health and infant nutrition.

Novel and significant findings: The article reveals the delicate differences of protein profiles of BC, Milk-L1 and Milk-L2, which have not been comprehensively studied before. The article shows that BC has higher levels of immune proteins, such as immunoglobulins and lactoferrin, than mature milk. The article also demonstrates that Milk-L1 has higher levels of anti-microbial proteins, such as β-defensins and bovine hemoglobin subunit, than Milk-L2, while Milk-L2 has higher levels of enteric β-defensin, sterol regulatory element binding transcription factor 1, sydecan-2 and cysteine-rich secretory protein 2 than Milk-L1.

Implications for infant formula development: The article provides a valuable insight into the distinct proteomic profiles of BC, Milk-L1 and Milk-L2, which can help to optimize the protein ingredients for infant formula. The article suggests that BC may be more suitable for preterm infants who need more immune protection, while Milk-L1 may be more beneficial for term infants who need more anti-microbial defense. The article also indicates that Milk-L2 may have some advantages for intestinal health and lipid metabolism. It will be interesting to mention that goat milk is allowed to use in formula too in some regions of the world.

Author Response

Dear editor

Thank you so much for considering our work for publication in Foods. We appreciate all the reviewer’s comments and suggestions.

To the reviewer: Thank you very much for your positive evaluation of this manuscript. We have included the suggested information on the trend of goat milk supplementation in infant formula in the discussion section, by stating that "Mature milk, both bovine and goat milk, also play significant roles in infant nutrition and are extensively used as the protein bases for infant formula." Please see Line 263.

Best regards,

Yuhui Ye, PhD

Jiangsu Academy of Agricultural Sciences